# Effect of Dietary Supplements with ω-3 Fatty Acids, Ascorbic Acid, and Polyphenolic Antioxidant Flavonoid on Gene Expression, Organ Failure, and Mortality in Endotoxemia-Induced Septic Rats

**DOI:** 10.3390/antiox12030659

**Published:** 2023-03-07

**Authors:** Yolanda Prado, Cesar Echeverría, Carmen G. Feijóo, Claudia A. Riedel, Claudio Cabello-Verrugio, Juan F. Santibanez, Felipe Simon

**Affiliations:** 1Laboratory of Integrative Physiopathology, Faculty of Life Sciences, Universidad Andres Bello, Santiago 8370186, Chile; 2Millennium Institute on Immunology and Immunotherapy, Santiago 8331150, Chile; 3Laboratory of Molecular Biology, Nanomedicine and Genomics, Faculty of Medicine, University of Atacama, Copiapo 1532502, Chile; 4Fish Immunology Laboratory, Faculty of Life Sciences, Universidad Andres Bello, Santiago 8370186, Chile; 5Laboratory of Endocrinology-Immunology, Faculty of Life Sciences, Universidad Andres Bello, Santiago 8370186, Chile; 6Laboratory of Muscle Pathology, Fragility and Aging, Faculty of Life Sciences, Universidad Andres Bello, Santiago 8370186, Chile; 7Center for the Development of Nanoscience and Nanotechnology (CEDENNA), Universidad de Santiago de Chile, Santiago 8350709, Chile; 8Institute for Medical Research, National Institute of the Republic of Serbia, University of Belgrade, 11129 Belgrade, Serbia; 9Integrative Center for Biology and Applied Chemistry (CIBQA), Bernardo O’Higgins University, Santiago 8370993, Chile; 10Millennium Nucleus of Ion Channel-Associated Diseases, Santiago 8380453, Chile

**Keywords:** food supplements, fatty acids, polyphenolic antioxidant flavonoids, endothelium, fibrosis, hyperpermeability, organ function, mortality, sepsis

## Abstract

Sepsis syndrome develops through enhanced secretion of pro-inflammatory cytokines and the generation of reactive oxygen species (ROS). Sepsis syndrome is characterized by vascular hyperpermeability, hypotension, multiple organ dysfunction syndrome (MODS), and increased mortality, among others. Endotoxemia-derived sepsis is an important cause of sepsis syndrome. During endotoxemia, circulating endotoxin interacts with endothelial cells (ECs), inducing detrimental effects on endothelium function. The endotoxin induces the conversion of ECs into fibroblasts, which are characterized by a massive change in the endothelial gene-expression pattern. This downregulates the endothelial markers and upregulates fibrotic proteins, mesenchymal transcription factors, and extracellular matrix proteins, producing endothelial fibrosis. Sepsis progression is modulated by the consumption of specific nutrients, including ω-3 fatty acids, ascorbic acid, and polyphenolic antioxidant flavonoids. However, the underlying mechanism is poorly described. The notion that gene expression is modulated during inflammatory conditions by nutrient consumption has been reported. However, it is not known whether nutrient consumption modulates the fibrotic endothelial gene-expression pattern during sepsis as a mechanism to decrease vascular hyperpermeability, hypotension, MODS, and mortality. Therefore, the aim of this study was to investigate the impact of the consumption of dietary ω-3 fatty acids, ascorbic acid, and polyphenolic antioxidant flavonoid supplements on the modulation of fibrotic endothelial gene-expression patterns during sepsis and to determine the effects on sepsis outcomes. Our results indicate that the consumption of supplements based on ω-3 fatty acids and polyphenolic antioxidant flavonoids was effective for improving endotoxemia outcomes through prophylactic ingestion and therapeutic usage. Thus, our findings indicated that specific nutrient consumption improves sepsis outcomes and should be considered in treatment.

## 1. Introduction

Sepsis syndrome is the most prevalent cause of mortality worldwide in critically ill patients admitted to intensive care units (ICU) [1,2]. Sepsis syndrome develops through an overactivation of the immune system, which involves the enhanced secretion of pro-inflammatory cytokines and the generation of reactive oxygen species (ROS) [3,4]. Sepsis syndrome is characterized by decreased blood pressure (hypotension), multiple organ dysfunction syndrome (MODS), and increased mortality, among others [5,6]. Despite numerous basic and clinical studies addressing sepsis syndrome, current therapies for treating it and its sequelae are unsatisfactory [4,5,6]. Endotoxemia-derived sepsis is an important cause of sepsis syndrome and is frequently characterized by the deposition of large amounts of the Gram-negative bacterial endotoxin lipopolysaccharide (LPS) [7,8]. During endotoxemia, the endotoxin circulating in the bloodstream interacts with the endothelial cells (ECs) located in the internal endothelial monolayer of blood vessels, inducing detrimental effects on endothelium function [8,9,10,11].

It is well accepted that endothelial dysfunction is an important factor in the pathogenesis of endotoxemia-derived sepsis syndrome and other inflammatory diseases. Endotoxin is able to induce the conversion of ECs into fibroblasts [12,13,14]. Endotoxin-induced endothelial fibrosis is mediated through a process known as endothelial-to-mesenchymal transition (EndMT) in a similar way to that observed using the best-studied EndMT inducer, tumor growth factor β (TGF-β) [15,16,17,18,19]. Endothelial fibrosis has been detected in septic ICU patients and in several animal septic models, and the pharmacological inhibition of endotoxemia-induced endothelial fibrosis improves organ failure and survival [13,20,21]. Endotoxemia-induced endothelial fibrosis is characterized by a massive change in the endothelial gene-expression profile, which downregulates endothelial markers, including vascular endothelial cadherin (VE-cadherin) and platelet endothelial cell adhesion molecule 1 (PECAM-1). This change also strongly upregulates several other genes, including fibrotic and extracellular matrix (ECM) proteins such as the α۔smooth muscle actin (α-SMA), fibroblast-specific protein 1 (FSP-1), collagen type I and III (Col I/III), and fibronectin (FN), as well as transcription factors such as nuclear factor kappa-light-chain-enhancer of activated B cells (NF-κB), Slug, and Twist. It also upregulates pro-inflammatory cytokines such as tumor necrosis factor α (TNF-α), interleukin-1β (IL-1β), and IL-6, as well as oxidative enzymes such as NAD(P)H oxidase (NOX), among several others [15,16,22,23,24,25,26]. Endothelial fibrosis has been associated with the origin and progression of several features of sepsis syndrome, including endothelial monolayer disruption, which promotes vascular hyperpermeability and edema formation, endothelial migration, refractory hypotension, pro-inflammatory cytokine production, MODS, and increased mortality [9,13,20,23,27,28,29,30].

It has been reported that specific nutrient consumption modulates sepsis. The dietary intake of ω-3 fatty acids enhances survival in septic mice, improves sepsis-induced intestinal inflammation and oxidative stress injury, and reduces the mortality rate of sepsis-induced acute respiratory distress syndrome [31,32,33]. Furthermore, dietary ascorbic acid decreases mortality and suppresses the inflammatory response to endotoxemia and sepsis, showing potential action on ECs [34,35]. Moreover, polyphenolic antioxidant flavonoid consumption improves endotoxemia and septic condition. Several types of flavonoids improve MODS and increase survival against sepsis [36,37,38]. The evidence suggests that the nutrient composition of food affects sepsis progression and its outcomes, but the molecular underlying mechanism has not been completely described.

It has been reported that gene expression is modulated under inflammatory conditions by nutrient consumption. The consumption of ω-3 fatty acids reduces inflammation by downregulating pro-inflammatory cytokines TNF-α, IL-1β, and IL-6 [39,40]. Ascorbic acid has the capacity to modulate gene expression, including collagen and fibronectin genes, in response to several stimuli, such as oxidative stress [41,42,43,44]. Interestingly, ascorbic acid regulates extracellular matrix production through collagen and fibronectin gene expression [44,45]. Furthermore, pro-inflammatory cytokine expression decreases as a consequence of flavonoid supplementation [46]. However, it is not known whether specific nutrient consumption modulates the fibrotic endothelial gene-expression pattern during sepsis to inhibit sepsis-induced endothelial fibrosis as a mechanism to decrease vascular endothelial hyperpermeability, hypotension, MODS, and mortality.

Therefore, the aim of this study was to investigate the effect of dietary supplements with ω-3 fatty acids, ascorbic acid, and polyphenolic antioxidant flavonoid on the modulation of fibrotic endothelial gene-expression patterns during sepsis and its effects on improving vascular hyperpermeability, hypotension, MODS, and mortality.

## 2. Materials and Methods

### 2.1. Animals, Dietary Supplement Administration, Endotoxemia Induction, and Parameter Recordings

Male Sprague Dawley (SD) rats weighing from 190 to 210 g were used. They were housed in cages with water and food ad libitum, a 12 h light/dark cycle, and 25 ± 1 °C temperature. All experimental procedures in rats were approved by the Commission of Bioethics and Biosafety of Universidad Andres Bello (N°002/2020) and following instruction form the Guide for the Care and Use of Laboratory Animals from the National Research Council and the American Association for Laboratory Animal Science. This research complies with the commonly-accepted 3Rs.

Rats were treated with 0.9% NaCl saline solution (saline-treated condition) or with the endotoxin lipopolysaccharide from *E. coli* (LPS (Sigma-Aldrich, Saint Louis, MO, USA), 10 mg/kg) (endotoxemia-treated condition) by intraperitoneal (I.P.) injection (50 µL), and animals were followed during the experiment. Saline-treated and endotoxemia-treated rats were subjected to dietary supplementation by means of a prophylactic (4 weeks before and during the 7 days of endotoxemia) and a therapeutic (only during the 7 days of endotoxemia) protocol. The period of prophylactic and therapeutic supplement treatments is compatible with those previously reported [31,47,48,49]. Furthermore, the period for endotoxemia induction is similar to those reported by us and other groups [3,13,30,50,51,52,53].

The composition of the dietary supplementation was as follows: sham supplementation (Sham): 2 mL water; ω-3 fatty acid supplementation (ω-3 FA): 300 mg ω-3 FAs (Sigma-Aldrich) (composed by 180 mg EPA and 120 mg DHA in 1000 mg (1.075 mL) of fish oil); ascorbic acid supplementation (AsA): 600 mg of AsA (Sigma-Aldrich) (2 mL of 300 mg/mL AsA solution in water); flavonoid supplementation (Flav): 6 mg of mixed flavonols, flavanones, isoflavones, flavones, flavan-3-ols, and anthocyanins, which were administered in a 2 mL of total solution (water or oil solvent) (all from Sigma-Aldrich). The dietary supplementation was administered daily by gavage. Twelve rats per group were used.

Rat blood samples were collected in a sodium citrate or lithium heparin-containing blood collection Vacutainer^®^ tube (Becton Dickinson Co., East Rutherford, NJ, USA) through tail-vein puncture or cardiac puncture after anesthetization. The obtained blood was immediately centrifuged at 4000 rpm for 10 min at 4 °C to separate the plasma and immediately stored at −80 °C. Systolic blood pressure (P_S_) and heart rate (H_R_) were acquired with a physiological recording system and a pressure tail cuff for a noninvasive blood-pressure-recording system for rats (ML125/R), coupled with an MLT125/R pulse transducer (AD Instruments, Bella Vista, Australia). To perform the recordings of P_S_ and H_R_, animals were recorded for 1 min several times to obtain reliable values. All transducers were connected to a PowerLab^®^ 8/30 (AD Instruments), and physiological variables were instantaneously displayed through Chart^®^ software (v6.1.1, AD Instruments).

### 2.2. Primary Rat Mesenteric Endothelial Cell (RMEC) Isolation and mRNA Expression Determination by RT-qPCR

Primary rat mesenteric endothelial cells (RMECs) were isolated from the mesenteric artery. The mesenteric artery was occluded on its distal end and cannulated from its proximal end with a polyethylene tubing connected to a 21-gauge syringe. The mesentery was surgically removed and washed with sterile PBS. For the enzymatic isolation of RMECs, the mesenteric artery was slowly perfused in a culture hood for 5 min with 5 mL M-199 medium supplemented with 40 μL penicillin/streptomycin (10,000 U/mL/10,000 μg/mL), 20 μL Fungizone (250 μg/mL), and 12.5 mg collagenase type II. The cell suspension was centrifuged at 3000 rpm for 7 min; the pellet was reconstituted in 3 mL M-199 medium supplemented with 8 mL/L of Pen/Strep (10,000 U/mL/10,000 μg/mL), 4 mL/L of Fungizone (250 μg/mL), 10% FBS, and 10% CCS. Thereafter, RMECs were subjected immediately to mRNA expression determination. RT-qPCR experiments were performed to measure mRNA expression levels in RMECs. Total RNA was extracted with Trizol according to the manufacturer’s protocol (Invitrogen, Carlsbad, CA, USA). DNAse I-treated RNA was used for reverse transcription using the Super Script II Kit (Invitrogen). Equal amounts of RNA were used as templates in each reaction. Quantitative-PCR was performed using the SYBR Green PCR Master Mix (AB Applied Biosystems, Foster City, CA, USA). Assays were run using a RotorGene instrument (Corbet Research, Sydney, Australia). The determination of mRNA expression was normalized relative to 28S mRNA and were expressed relative to sham-supplemented condition.

### 2.3. In Vivo Permeability Assay and MODS Determination

At the end of experiments, rats were anesthetized with isoflurane and injected with Evans blue dye (EBD (Sigma-Aldrich), 80 mg/kg, intravenous) for 10 min. EBD binds to plasma proteins and extravasates to tissue parenchyma at sites of increased vascular permeability. Then, rats were euthanized and perfused with saline solution via the left ventricle to wash excess EBD. Mesentery, kidney, and liver were removed, washed in cold saline solution, gently dried using a paper towel, weighted, and prepared for EBD extraction. Briefly, organs were weighted and homogenized with 100% trichloroacetic acid in a 1:2 (mg:mL) proportion. Homogenates were centrifuged at 4180× *g* for 30 min, and the supernatant optical density was determined spectrophotometrically at 630 nm. To assess MODS, plasma levels of aspartate aminotransferase (AST), alanine aminotransferase (ALT), total bilirubin (TBIL), and gamma-glutamyl transferase (GGT) were quantified to measure liver function; creatinine (CRE) and blood urea nitrogen (BUN) were quantified to measure kidney function; lactic acid (LAC) and glycemia (GLY) were quantified to measure metabolic function. MODS markers were quantified using a Piccolo Xpress Chemistry Analyzer equipped with General Chemistry 13, MetLyte Plus CRP, and Basic Metabolic Panel Plus panels (Abaxis, Union City, CA, USA), according to the manufacturers’ instructions.

### 2.4. Statistical Analyses

Differences were considered significant at *p* < 0.05. Statistical differences were assessed by the Mann–Whitney U-test and one-way analysis of variance (one-way ANOVA) (or Kruskal–Walli’s type), followed by Dunn’s post hoc test and two-way analysis of variance (two-way ANOVA) followed by Tukey post hoc test. See the figure legends for the specific test used. Survival Kaplan–Meier curves were compared by the log-rank (Mantel–Cox) test and the Gehan–Breslow–Wilcoxon test to determine survival rates. Contingency analyses with Fisher’s exact test were used to assess the relative risk of death. Statistical testing was two-sided and used the 5% significance level. The data were analyzed with GraphPad Prism version 9.4 (GraphPad Software, LLC, Boston, MA, USA). Samples used in the study were defined to identify the mean magnitude effect of a ≥2-fold change with standard deviations of 10%. Accordingly, a sample size of 12 rats per group would provide 90% statistical power using a two-sided 0.05 significance level.

## 3. Results

### 3.1. Impact in EndMT Gene Expression through Dietary Supplementation Based on ω-3 FA, Ascorbic Acid, or Polyphenolic Antioxidant Flavonoids in Endotoxemic Rats

To determine the effects on EndMT-related gene expression and sepsis syndrome, rats were subjected to saline treatment or endotoxemia and fed with ω-3 fatty acid, ascorbic acid, or flavonoid supplementation by means of a prophylactic protocol (4 weeks before and for 7 days of endotoxemia) or therapeutic protocol (only for the 7 days of endotoxemia) (Figure 1A). Firstly, we established the change in endotoxemia-induced mRNA expression profiles for several selected genes involved in EndMT-mediated endothelial fibrosis in endotoxemic rats. For this purpose, we analyzed the mesenteric endothelium expression profile because it is a vascular network that frequently undergoes vascular disease associated with fibrosis and regulates a relevant portion of total blood flow. The mesentery is a vascular area that represents blood vessels in other tissues well. Thus, ECs from the mesentery are an appropriate model for studying dysfunctional endothelial changes and are useful as representatives of the endothelium from internal organs.

Primary rat mesenteric endothelial cells (RMECs) were extracted from the rats of all groups (see Figure 1A), and mRNA expression was immediately determined to avoid any possible cell transformation. The results showed that RMECs extracted from endotoxemia-treated rats of all groups exhibited decreased mRNA expression of the endothelial markers VE-Cadherin, PECAM-1, von Willebrand Factor (vWF), and collagen type IV (Col IV) compared to saline-treated rats. However, endotoxemic rats showed increased mRNA expression of the fibrotic and ECM markers alpha smooth muscle actin (α-SMA), smooth muscle protein 22 alpha (SM22α), FSP-1, Col I, Col III, and FN. Furthermore, the mRNA expressions of the EndMT inducers IL-1β, IL-6, TNF-α, transforming growth factor beta-1 (TGF-β1), and transforming growth factor beta-2 (TGF-β2) were also increased in endotoxemic rats. The mRNA expression of the cytokine receptors IL-1R, IL-6R, TNFR, and TβRI were also increased in endotoxemic rats, while TβRI did not show changes compared to saline-treated rats. All transcription factors tested (NF-κB, Smad4, Slug, Twist, Snail, and Zeb-1) showed increased mRNA expression in endotoxemic rats. In addition, mRNA expression of NOX-2 from the endotoxemic condition was increased, whereas mRNA expression of NOX-1 and NOX-4 did not show changes compared to saline-treated rats (Figure 1B). The survival percentage of the endotoxemic rats was 25% after 7 days of endotoxemia challenge (Figure 1C).

Interestingly, the endotoxemic rats fed with a diet supplemented with ω-3 fatty acid following the therapeutic and prophylactic protocols showed a reduced decrease in mRNA expression of the endothelial markers (Figure 2A,B, left panels). Furthermore, the ω-3 fatty acid-supplemented endotoxemic rats in the therapeutic and prophylactic protocols showed a reduced increase in mRNA expression of the fibrotic and ECM markers, EndMT inducers, cytokine receptors, transcription factors, and NOX-2 enzyme compared with saline-treated rats (Figure 2A,B, middle and right panels). Notably, the ω-3 fatty acid-supplemented endotoxemic rats in the therapeutic and prophylactic protocol showed a non-significant change in mRNA expression of several EndMT-related genes compared with saline-treated rats (Figure 2A,B), suggesting that the ω-3 fatty acid supplementation efficiently abolished the endotoxemia-induced change in the EndMT expression profile. Similar results were obtained when the supplementation with ascorbic acid (AsA) was used (Figure 2C,D). However, because the non-significant change was almost not detected, it is suggested that AsA supplementation has a minor impact on EndMT-related mRNA expression compared with ω-3 fatty acid supplementation.

Furthermore, similar results to those observed with ω-3 fatty acid supplementation were obtained when the supplementation with flavonoids was assessed (Figure 2E,F). However, a non-significant change in mRNA expression was observed in only the prophylactic treatment of flavonoid supplementation. These results indicate that predominantly in the prophylactic protocol, the ω-3 fatty acid and flavonoid supplementation exhibited similar and strong protection against the endotoxemia-induced EndMT gene-expression change.

Notably, the survival percentages in the endotoxemic rats fed with a diet supplemented with ω-3 fatty acid in the therapeutic and prophylactic protocol were 50% and 83%, respectively, (Figure 2G,H, respectively). This shows protection against endotoxemia compared with the sham-supplemented endotoxemic rats after 7 days of endotoxemic challenge. Supplementation with AsA produced a modest increase in survival (25% to 33%) in only the prophylactic treatment, whereas the therapeutic treatment showed the same survival percentage as that observed in the sham-supplemented endotoxemic rats (Figure 2I,J, respectively). Importantly, the survival percentages in the flavonoid-supplemented endotoxemic rats in the therapeutic and prophylactic protocols were 33% and 50%, respectively, after 7 days of endotoxemia challenge (Figure 2K,L, respectively). This shows protection against endotoxemia compared with the sham-supplemented endotoxemic rats after 7 days of endotoxemic challenge. These results indicated that principally, the ω-3 fatty acid and flavonoid supplementation produced protection against mortality induced by endotoxemia.

### 3.2. Effects on Vascular Hyperpermeability through Dietary Supplementation Based on ω-3 FA, Ascorbic Acid, or Polyphenolic Antioxidant Flavonoids in Endotoxemic Rats

EndMT-induced endothelial fibrosis produces a severe disruption of the vascular endothelium monolayer, which promotes increased permeability and generates edema formation. Considering this, we tested whether the consumption of a diet supplemented with ω-3 fatty acid, ascorbic acid, or flavonoids is able to improve endotoxemia-induced vascular hyperpermeability. To this end, blood vessel hyperpermeability was tested in the mesentery and in two additional relevant organs that are severely affected during sepsis: the liver and the kidney. This was performed using endotoxemic rats by means of the Evans blue dye (EBD) permeability assay.

The mesentery, liver, and kidney extracted from sham-treated endotoxemic rats showed increased EBD accumulation, which indicates that blood vessels leaked to the interstitial compartment in contrast with saline-treated rats (Figure 3A–C, left panel). Interestingly, the ω-3 fatty acid-supplemented endotoxemic rats in the therapeutic protocol and notably in the prophylactic protocol showed a strong decrease in the endotoxemia-induced hyperpermeability in the mesentery, liver, and kidney (Figure 3A–C, middle left panel). AsA-supplemented endotoxemic rats showed decreased endotoxemia-induced hyperpermeability in the mesentery and liver, but not in the kidney (Figure 3A–C, middle right panel) with the prophylactic protocol. The therapeutic protocol did not show any change in endotoxemia-induced hyperpermeability in all tested organs. The flavonoid-supplemented endotoxemic rats in only the prophylactic protocol showed decreased endotoxemia-induced hyperpermeability in all tested organs (Figure 3A–C, right panel).

During endotoxemia, vascular hyperpermeability is associated with changes in EndMT gene expression, so correlation analyses were performed between these two factors. The results showed that in the endotoxemic condition, the increased permeability in the mesentery (Appendix A), liver (Appendix A), and kidney (Appendix A) correlates with the change in mRNA expression of most of the EndMT-related genes tested in Figure 2. Interestingly, in all tested organs, VE-Cadherin, α-SMA, Col III, TNF-α, and NF-κB showed correlation coefficients (r^2^) and *p*-values (*p*) higher than 0.8 and 0.01, respectively. The expression of these genes at the protein level was verified and showed concordant results compared to those shown in the mRNA determination (Appendix A).

Interestingly, higher changes in mRNA expression of VE-Cad, α-SMA, Col III, TNF-α, and NF-κB were found mainly in the non-surviving endotoxemic rats compared to survivors, as observed in the sham group (Figure 4A–E, left panel, respectively). The ω-3 fatty acid-supplemented endotoxemic rats had decreased hyperpermeability values in both the therapeutic and prophylactic protocols in non-survivors and survivors, compared to the corresponding condition in the sham group, showing higher changes in mRNA expression in the non-surviving ω-3 fatty acid-supplemented endotoxemic rats compared to survivors (Figure 4A–E, middle left panel). However, AsA-supplemented endotoxemic rats showed, to a major extent, non-significant differences in mRNA expression changes between non-surviving and surviving rats (Figure 4A–E, middle right panel). The flavonoid-supplemented endotoxemic rats showed higher changes in mRNA expression in the non-surviving rats subjected to the prophylactic protocol, whereas in the therapeutic protocol, non-significant differences were observed (Figure 4A–E, right panel).

Vascular hyperpermeability has detrimental effects on the function of several organs and tissues and promotes mortality, and it is increased during sepsis. As we expected, higher permeability values in the mesentery, liver, and kidney were found in the non-surviving endotoxemic rats compared to survivors, as observed in the sham group (Figure 5A–C, left panel). The ω-3 fatty acid-supplemented endotoxemic rats in the therapeutic and prophylactic protocols had decreased hyperpermeability values and showed the same distribution in the mesentery, liver, and kidney (Figure 5A–C, middle left panel). Higher permeability values were found in the non-surviving ω-3 fatty acid-supplemented endotoxemic rats compared to survivors. Interestingly, AsA-supplemented endotoxemic rats lost this distribution and showed non-significant differences in the permeability values between non-surviving and surviving rats (Figure 5A–C, middle right panel). The flavonoid-supplemented endotoxemic rats showed higher permeability values in the non-surviving rats in the prophylactic protocol, whereas in the therapeutic protocol, non-significant differences were observed (Figure 5A–C, right panel).

### 3.3. Effects on Endotoxemia-Induced Hypotension through Dietary Supplementation Based on ω-3 FA, Ascorbic Acid, or Polyphenolic Antioxidant Flavonoids in Endotoxemic Rats

Septic syndrome and endotoxemia are characterized by decreased blood pressure or hypotension. EndMT-induced increased hyperpermeability contributes to hypotension. Thus, we tested whether the consumption of a diet supplemented with ω-3 fatty acid, ascorbic acid, or flavonoids is able to improve endotoxemia-induced hypotension. Systolic pressure monitoring showed that the ω-3 fatty acid-supplemented endotoxemic rats in the therapeutic and prophylactic protocols showed strong improvement of hypotension (Figure 6A, middle left panel) compared to sham-supplemented endotoxemic rats (Figure 6A, left panel). However, AsA-supplemented endotoxemic rats did not show any change in endotoxemia-induced hypotension (Figure 6A, middle right panel). The flavonoid-supplemented endotoxemic rats showed improved hypotension when subjected to the prophylactic protocol, whereas in the therapeutic protocol, no change was detected (Figure 6A, right panel). As compensation for the hypotension, the heart rate was found to be increased in endotoxemia-treated rats (Figure 6B, left panel). The ω-3 fatty acid-supplemented endotoxemic rats in the therapeutic and prophylactic protocols showed decreased tachycardia (Figure 6B, middle left panel). However, AsA-supplemented endotoxemic rats did not show any change in tachycardia (Figure 6B, middle right panel). The flavonoid-supplemented endotoxemic rats showed improved tachycardia in rats subjected to the prophylactic protocol but not in those subjected to the therapeutic protocol (Figure 6B, right panel).

### 3.4. Effects on Endotoxemia-Induced MODS through Dietary Supplementation Based on ω-3 FA, Ascorbic Acid, or Polyphenolic Antioxidant Flavonoids in Endotoxemic Rats

MODS is a crucial factor during sepsis syndrome and severely impacts survival, which is promoted by hyperpermeability. Thus, we assessed whether the consumption of a diet supplemented with ω-3 fatty acid, ascorbic acid, or flavonoids is able to improve the endotoxemia-induced MODS by measuring blood markers of liver, renal, and metabolic function.

The ω-3 fatty acid-supplemented endotoxemic rats in the therapeutic and prophylactic protocols showed improved liver function with decreases in the endotoxemia-induced increased blood levels of the liver dysfunction markers alanine aminotransferase (ALT), aspartate aminotransferase (AST), total bilirubin (TBIL), and gamma-glutamyl transferase (GGT) (Figure 7A–D, middle left panel), compared to sham-supplemented endotoxemic rats (Figure 7A–D, left panel). AsA-supplemented endotoxemic rats did not show protection against endotoxemia-induced liver dysfunction (Figure 7A–D, middle right panel). The flavonoid-supplemented endotoxemic rats showed protection against liver dysfunction with decreases in endotoxemia-induced increased blood levels of ALT, AST, TBIL, and GGT in rats subjected to the prophylactic protocol, whereas in the therapeutic protocol, no change was detected (Figure 7A–D, right panel).

Regarding renal function, the ω-3 fatty acid-supplemented endotoxemic rats in the therapeutic and prophylactic protocols showed improved renal dysfunction with decreased blood levels of markers of renal dysfunction creatinine (CRE) and blood urea nitrogen (BUN) (Figure 7E,F, middle left panel) compared to sham-supplemented endotoxemic rats (Figure 7E,F, left panel). Furthermore, the BUN/creatinine ratio, which is an indicator of kidney disease risk, was increased in sham-supplemented endotoxemic rats (Figure 7G, left panel) but was limited in the ω-3 fatty acid-supplemented endotoxemic rats in the therapeutic and prophylactic protocols (Figure 7G, middle left panel). AsA and flavonoid-supplemented endotoxemic rats did not show modification in the BUN/creatinine ratio (Figure 7G, middle right and right panel). We also estimated GFR, which is a typical diagnostic test for chronic kidney disease, by using a validated plasma creatinine and urea-based equation to determine the estimated GFR (eGFR) in male Sprague Dawley rats [54]. The ω-3 fatty acid-supplemented endotoxemic rats in the therapeutic and prophylactic protocols showed improved eGFR (Figure 7H, middle left panel) compared to sham-supplemented endotoxemic rats (Figure 7H, left panel). However, AsA and flavonoid-supplemented endotoxemic rats did not show a change in eGFR (Figure 7H, middle right and right panel).

Next, we investigated the action of nutrient supplementation on endotoxemia-induced metabolic dysfunction. The ω-3 fatty acid-supplemented endotoxemic rats in the therapeutic and prophylactic protocols showed improved metabolic dysfunction. Endotoxemia-induced changes in the blood levels of lactic acid (LAC) and glycemia (GLY), which are markers of metabolic function (Figure 7I,J, middle left panel), were normalized compared to sham-supplemented endotoxemic rats (Figure 6I,J, left panel). However, AsA and flavonoid-supplemented endotoxemic rats did not show protection against endotoxemia-induced metabolic dysfunction (Figure 7I,J, middle right and right panel).

These results indicate that ω-3 fatty acid supplementation and, to a minor extent, flavonoid supplementation are able to preserve organ function during endotoxemia.

### 3.5. Impact on Survival and Risk of Death through Dietary Supplementation Based on ω-3 FA, Ascorbic Acid, or Polyphenolic Antioxidant Flavonoids in Endotoxemic Rats

To determine whether the consumption of a diet supplemented with ω-3 fatty acid, ascorbic acid, or flavonoids is able to improve survival and risk of death in endotoxemic rats, we analyzed survival curves and contingency analysis.

The ω-3 fatty acid-supplemented endotoxemic rats in the therapeutic and prophylactic protocols showed a significant difference compared with the endotoxemia-treated rats, as indicated by the log-rank (Mantel-Cox) test (Figure 8A). To give more weight to deaths at early time points, the Gehan–Breslow–Wilcoxon test was performed, which also showed that the ω-3 fatty acid-supplemented endotoxemic rats in the therapeutic and prophylactic protocols had an increased death incidence (Figure 8A). However, AsA-supplemented endotoxemic rats did not show any change compared with the endotoxemia-treated rats, as indicated by the log-rank (Mantel–Cox) test and the Gehan–Breslow–Wilcoxon test (Figure 8B). The endotoxemic condition in flavonoid-supplemented rats showed a significant difference when compared with the endotoxemic condition in rats subjected to the prophylactic protocol but not in those subjected to the therapeutic protocol (Figure 8C). Next, the contingency analysis showed that the ω-3 fatty acid supplementation decreased the risk of death in endotoxemic rats in both the therapeutic and prophylactic protocols compared with the endotoxemia-treated rats (Figure 8D). However, AsA-supplemented endotoxemic rats did not show any change in the risk of death (Figure 8E). The flavonoid-supplemented endotoxemic rats showed a significant decrease in the risk of death in the endotoxemic rats subjected to the prophylactic protocol but not in those subjected to the therapeutic protocol (Figure 8F).

Taken together, our results showed that the consumption of supplements based on ω-3 fatty acids, ascorbic acid, and polyphenolic antioxidant flavonoids was effective for improving several parameters during endotoxemia outcome through prophylactic ingestion and therapeutic usage (Appendix A).

## 4. Discussion

The finding that nutrition can regulate gene expression in healthy and diseased states has been widely studied. Studies have shown that polyunsaturated fatty acids regulate the expression of several genes, including DNA-binding proteins, nutrient-sensitive transcription factors including NF-κB, ROS regulation enzymes, transporters, cell adhesion, and proliferation, among several others [55,56,57,58]. Furthermore, an antioxidant-rich diet also shows modulation of the gene expression in healthy and diseased conditions depending on the antioxidant type used and the oxidative status of the tissues [59,60,61]. Consumption of vitamins C, D, and E also regulates gene expression in several cell types, including immune cells, astrocytes, and osteoblasts [62,63,64,65]. Inflammation-related genes are also differentially expressed in response to a low-calorie diet [66].

Specific nutrient consumption has been associated with gene expression during sepsis and other inflammatory conditions, thus impacting the sepsis outcome. It has been reported that consumption of ω-3 fatty acid reduces expression levels of genes involved in inflammation [39,40,67]. Interestingly, the intake of ω-3 fatty acid regulates the expression of genes involved in membrane architecture that modify intracellular signaling [68]. The beneficial effects of ω-3 fatty acids are principally promoted by eicosapentaenoic acid (EPA) and docosahexaenoic acid (DHA). Treatment with EPA or DHA reduces monounsaturated fatty acids and increases the polyunsaturated fatty acids in primary myometrial and leiomyoma cells [68]. The present study used fish oil containing a combination of EPA and DHA, which had a concentration of 30% relative to the total volume. This natural concentration was effective in improving the outcomes of endotoxemia in rats in this study. This finding opens broad therapeutic possibilities to use higher concentrations of EPA and DHA during sepsis, but further studies are needed to explore this issue. Notably, therapeutical actions of ω-3 fatty acids can be applied and even potentiated by combining them with vitamins, antioxidants, and other elements to alleviate several pathological conditions [69,70]. There is evidence of a beneficial effect of a combination of statins and ω-3 fatty acids on platelet function and other hemostatic mechanisms, as well as the inhibition of the inflammatory process. However, a combined treatment of statins and ω-3 fatty acid on sepsis has not been evaluated yet [71], so further experiments are needed to study this possibility. Dietary supplementation with ω-3 fatty acids also improves survival against sepsis and its complications [31,32,33]. Dietary ω-3 fatty acids increase the survival of *Staphylococcus aureus*-induced sepsis by reversing the deleterious effect of a high-saturated-fat diet [72]. Acute pancreatitis may cause systemic inflammatory response syndrome, organ failure, and death. The intake of ω-3 fatty acids decreases the risk of organ failure in patients with acute pancreatitis [73]. Dietary ω-3 fatty acids reduced mortality and decreased prostaglandin E_2_ production in Kupffer cells in a rat model of sepsis [74]. Ascorbic acid has the capacity to modulate gene expression in response to several stimuli, such has oxidative stress [41,42,43]. Interestingly, ascorbic acid induces collagen gene expression in human fibroblasts [44]. Congruently, ascorbic acid is postulated as a key mediator of the extracellular matrix production in trabecular meshwork cells in the eye, mediating the expression of collagen type I, fibronectin, and laminin [45]. Dietary ascorbic acid shows beneficial actions against endotoxemia and sepsis [34,35]. In polymicrobial sepsis, ascorbic acid regulates hepatic vasoregulatory gene expression and reduces hepatic microvascular dysfunction [75]. Interestingly, a phase I trial of intravenous ascorbic acid infusion in patients with severe sepsis showed a positive impact on multiple organ failure, reduced proinflammation biomarkers, and attenuation of vascular endothelial injury [76]. Notably, monocytes challenged with endotoxin and treated with ascorbic acid showed decreased proinflammatory cytokine production [77]. Similar results showed protection against sepsis and acute lung injury associated with sepsis by means of a decrease in the proinflammatory response, improved alveolar fluid clearance, and prevention of sepsis-associated coagulation malfunctions [78]. A polymicrobial sepsis study showed that ascorbic acid infusion diminishes oxidative stress and lipid peroxidation, modulates hepatic vasoregulatory gene expression, and decreases hepatic microvascular dysfunction [75]. A diet containing ascorbic acid combined with hydrocortisone and thiamine improves sepsis features and consequences [79]. The utilization of combined hydrocortisone, ascorbic acid, and thiamine therapy appears to be safe and provides potential benefits for the management of sepsis and septic shock, but some aspects remain controversial since their actions seem to be modest in decreasing mortality [80,81,82]. The consumption of polyphenolic antioxidant flavonoids regulates gene expression. The intake of some flavonoids such as resveratrol and epicatechins has been shown to regulate the gene expression of proteins involved in insulin secretion from β-cells and glucose transport and insulin signaling in adipocytes [83,84]. Supplementation with the flavonoid epicatechin modulates gene-expression profiles of immune cells in humans [85]. The flavonoid polyphenol quercetin upregulates neuron gene expression [86]. The flavonoid rhamnetin decreases organ failure in Gram-negative bacterial sepsis in mice [87]. Similarly, the flavonoids luteolin and icariin increase survival against polymicrobial sepsis [36,37]. The dietary flavonoid fisetin extracted from berries protects against polymicrobial sepsis-induced inflammatory response and multiple organ dysfunction [38]. The flavonoid acacetin exerts protection against sepsis [88]. Oral consumption of the flavonoid kaempferol showed decreased acute lung injury in polymicrobial sepsis [89]. Considering that a combination of several of these flavonoids was used in this study, a massive amount of work remains to clarify the individual contributions of each type of flavonoid in sepsis-induced EndMT gene expression and the deleterious outcomes of sepsis. Importantly, we evaluated the action of dietary supplementation on the expression changes induced by endotoxemia in the mesenterial endothelium because it is a relevant vascular region to study fibrosis and pathological conditions linked to fibrosis [90,91,92]. Moreover, ECs from the mesentery are a suitable and interesting model for studying dysfunctional endothelial changes [93,94].

During sepsis, endothelial dysfunction emerges as a main factor to generate MODS. The endothelium regulates vascular functions such as vascular reactivity, hemostasis, and vascular permeability [95]. During sepsis, endothelial permeability is increased and produces vascular leakage to the interstitium, generating edema [20,29,50,96]. In edema formation, blood volume into the vasculature decreases and produces severe hypotension, which increases the risk of death [20,27,51,97]. Hypotension damages microvascular perfusion and generates massive MODS, which increases mortality and the risk of death [96,98,99]. Thus, the endotoxic-induced EndMT that triggers the endothelial gene-expression profile promotes the decreased expression of the endothelial adhesion proteins and the upregulation of the fibrotic and ECM proteins. This disrupts the endothelial architecture, contributes to the endothelial hyperpermeability, and causes MODS. Endothelial fibrosis has been detected in septic ICU patients, and the expression changes of VE۔cadherin have emerged as an accurate diagnostic tool. In endotoxemic rats, the pharmacological inhibition of endotoxemia-induced endothelial fibrosis improves MODS and survival [13,20,21].

The prophylactic and the therapeutic experimental models are both valuable in a clinical context. ICU patients not affected by sepsis are at risk of undergoing sepsis because ICU non-septic patients often progress into sepsis or septic shock. Prophylactic treatment could be useful in the prevention of the adverse effects of nosocomial infections, such as catheter-induced urinary tract infections, intra-hospital acquired pneumonia, skin infections (mainly in burn patients), bloodstream infections, and surgery-induced infections [100,101]. Enteral nutrition can prevent catabolic states observed in critical illness as well as in intestinal villi atrophy, apoptosis of enterocytes, dysbiosis, and alterations of immune function in the gut [102,103]. Conversely, ω-3 fatty acid supplementation or fish oil nutrition has been used via parenteral nutrition in the septic subject, improving sepsis outcomes [104,105,106]. Further studies are needed to determine the most effective path of administration for each supplement against sepsis.

Sepsis syndrome is the main cause of death in ICU patients, and the current therapy is unsatisfactory. Thus, the results demonstrating that nutrient supplementation inhibited endotoxemia-induced endothelial fibrosis, hypotension, MODS, and mortality during endotoxemia are of great importance. Therefore, specific nutrient consumption could be a therapeutic alternative against deleterious effects of sepsis and other endothelial fibrosis-mediated pathologies.

## 5. Conclusions

Our results indicate that the consumption of supplements based on ω-3 fatty acids, ascorbic acid, and polyphenolic antioxidant flavonoids can decrease the endotoxemia-induced gene-expression change associated with endothelial fibrosis. Furthermore, endotoxemia-induced mortality was decreased by the consumption of supplements based on ω-3 fatty acids, mainly via prophylactic ingestion as well as in therapeutic usage, and by the consumption of polyphenolic antioxidant flavonoids by means of prophylactic ingestion, while the consumption of supplement of ascorbic acid showed no significant difference. Thus, our findings indicated that specific nutrient consumption improves sepsis outcomes and should be considered in treatment against sepsis and other systemic inflammatory diseases.

## Figures and Tables

**Figure 1 antioxidants-12-00659-f001:**
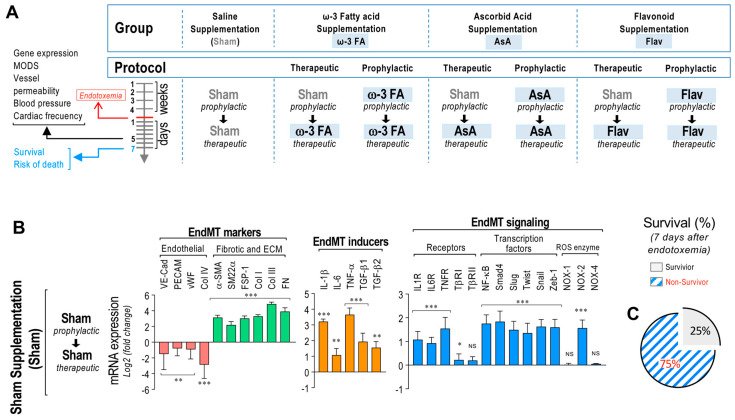
EndMT gene expression in endotoxemic rats. (**A**) Scheme of the strategy of saline– and endotoxemia–treated rats subjected to dietary supplementation by means of a prophylactic (4 weeks before and during the 7 days of endotoxemia) and a therapeutic (only during the 7 days of endotoxemia) protocol. Sham supplementation (Sham), ω–3 fatty acid supplementation (ω–3 FA)*,* ascorbic acid supplementation (AsA), and flavonoid supplementation (Flav). (**B**) The mRNA expression of EndMT markers (endothelial markers (red bars) and fibrotic and ECM markers (green bars)), EndMT inducers (orange bars), and EndMT signaling (blue bars) was detected in rats subjected to endotoxemia. (**C**) Survival percentage was determined 7 days after endotoxemia induction in rats subjected to endotoxemia. Statistical differences were assessed by a one-way analysis of variance (ANOVA) (Kruskal–Wallis) followed by Dunn’s post hoc test. * *p* < 0.05, ** *p* < 0.01, *** *p* < 0.001 compared with the saline-treated condition. NS: non-significant. Determination of mRNA expression was normalized relative to 28S mRNA and is expressed relative to sham-supplemented condition. Values are expressed as the mean ± SD. (*N* = 12).

**Figure 2 antioxidants-12-00659-f002:**
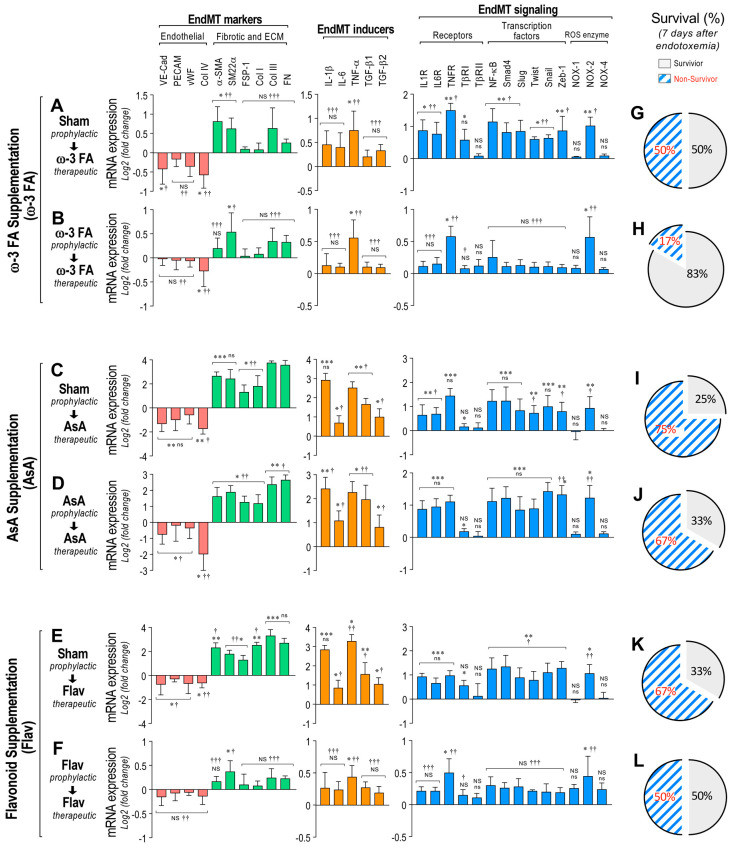
Impact on EndMT gene expression through dietary supplementation based on ω–3 FA, ascorbic acid, or polyphenolic antioxidant flavonoids in endotoxemic rats. (**A**–**D**) The mRNA expression of EndMT markers (endothelial markers (red bars) and fibrotic and ECM markers (green bars)), EndMT inducers (orange bars) and EndMT signaling (blue bars) were detected in rats subjected to endotoxemia and supplemented with ω–3 fatty acid (ω-3 FA, (**A**,**B**)), ascorbic acid (AsA, (**C**,**D**)), or polyphenolic antioxidant flavonoids (Flav, (**E**,**F**)) through a therapeutic (**A**,**C**,**E**) and a prophylactic (**B**,**D**,**F**) protocol. (**G**–**L**) Survival percentage was determined 7 days after endotoxemia induction in rats subjected to endotoxemia and supplemented with ω–3 fatty acid (ω–3 F (**G**,**H**)), ascorbic acid (AsA, (**I**,**J**)), or polyphenolic antioxidant flavonoids (Flav, (**K**,**L**)) through a therapeutic (**G**,**I**,**K**) and a prophylactic (**H**,**J**,**L**) protocol. Statistical differences were assessed by a one-way analysis of variance (ANOVA) (Kruskal–Wallis) followed by Dunn’s post hoc test. * *p* < 0.05, ** *p* < 0.01, *** *p* < 0.001 compared with the saline-treated condition. † *p* < 0.05, †† *p* < 0.01, ††† *p* < 0.001 compared with the mRNA-expression values from endotoxemia-treated rats shown in Figure 1B. NS: non-significant. Determination of mRNA expression was normalized relative to 28S mRNA and are expressed relative to sham-supplemented condition. Values are expressed as the mean ± SD. (*N* = 12).

**Figure 3 antioxidants-12-00659-f003:**
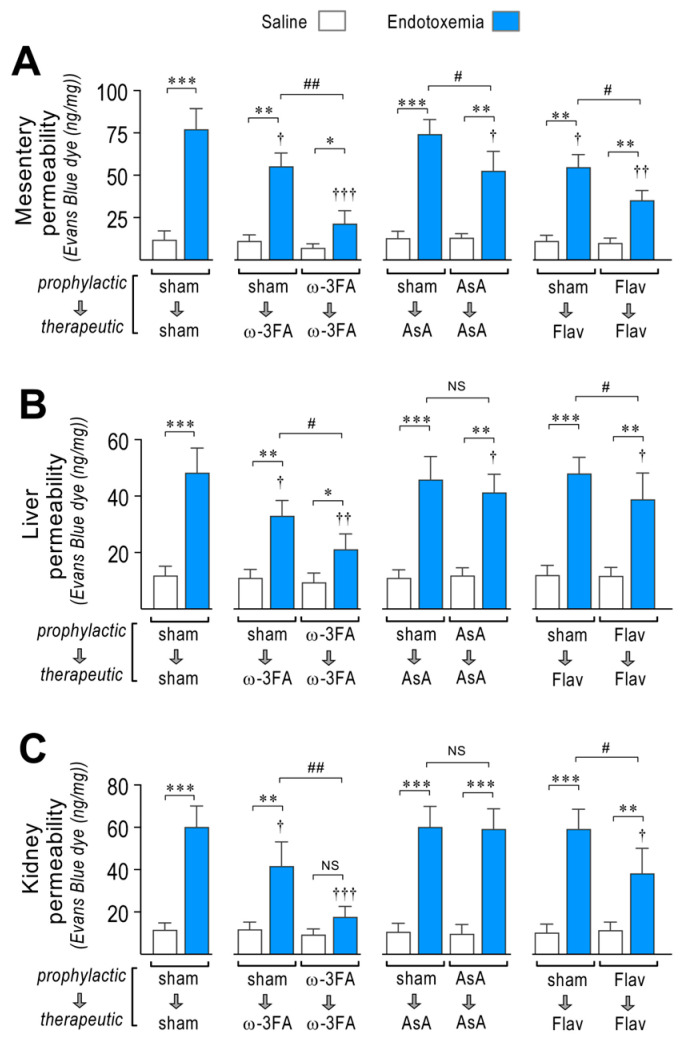
Effects in vascular hyperpermeability through dietary supplementation based on ω-3 FA, ascorbic acid, or polyphenolic antioxidant flavonoids in endotoxemic rats. After saline (open bars) and endotoxemia (blue bars) treatments that were supplemented with water (sham), ω-3 fatty acid (ω-3 FA), ascorbic acid (AsA), or polyphenolic antioxidant flavonoids (Flav), through a therapeutic and a prophylactic protocol, the mesentery (**A**), kidney (**B**), and liver (**C**) were harvested and processed for EBD extraction to determine blood vessel hyperpermeability. EBD content was normalized to total sample weight. Statistical differences were assessed by two-way ANOVA followed by Dunnett’s post hoc test. * *p* < 0.05; ** *p* < 0.01; *** *p* < 0.001 comparing saline and endotoxemia condition. † *p* < 0.05, †† *p* < 0.01, ††† *p* < 0.001 compared with sham-treated condition. # *p* < 0.05; ## *p* < 0.01 comparing non-survivor endotoxic condition between therapeutic and prophylactic protocol. NS: non-significant. Results are expressed as the mean ± SD (*N* = 12).

**Figure 4 antioxidants-12-00659-f004:**
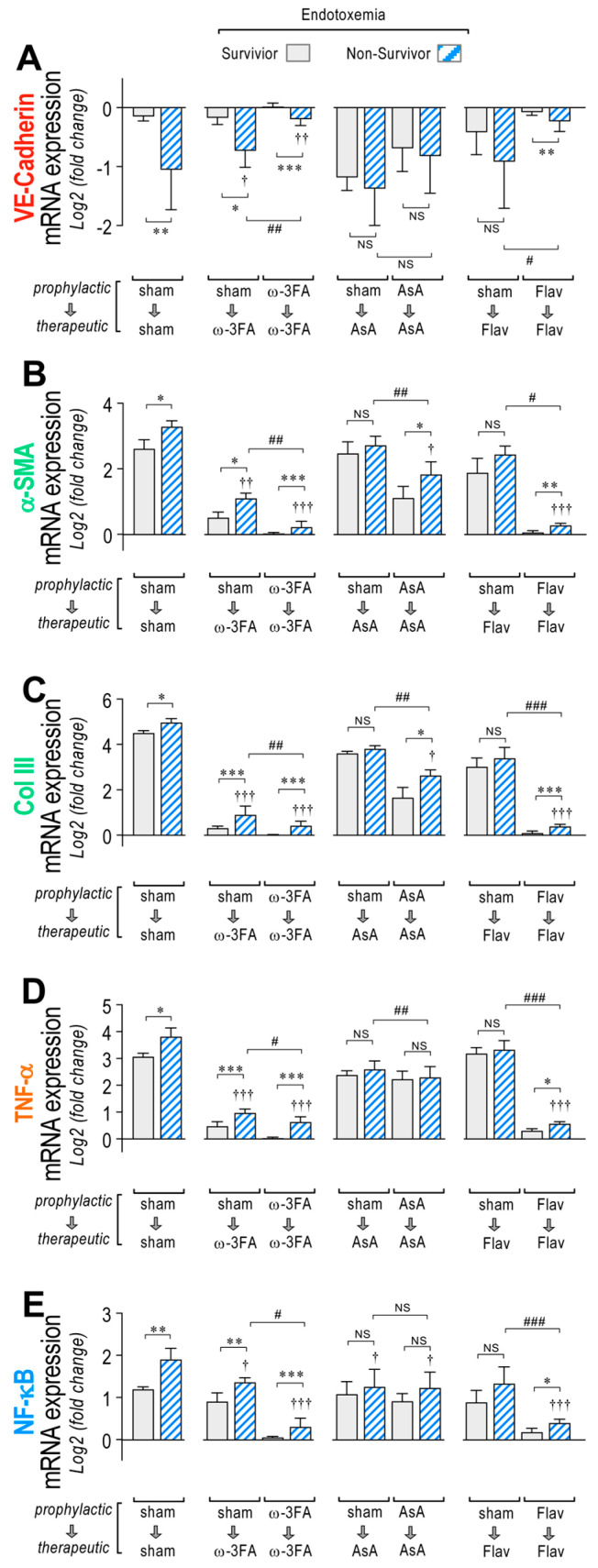
VE-Cadherin, α-SMA, Col III, TNF-α, and NF-κB mRNA expression through dietary supplementation based on ω-3 FA, ascorbic acid, or polyphenolic antioxidant flavonoids in survivor and non-survivor endotoxemic rats. After endotoxemia treatment supplemented with water (sham), ω-3 fatty acid (ω-3 FA), ascorbic acid (AsA), or polyphenolic antioxidant flavonoids (Flav), through a therapeutic and a prophylactic protocol, the mRNA expression of VE-Cadherin (**A**), α-SMA (**B**), Col III (**C**), TNF-α (**D**), and NF-κB (**E**) were determined in survivor (grey bars) and non-survivor (dashed bars) endotoxemic rats. Statistical differences were assessed by two-way ANOVA followed by Dunnett’s post hoc test. * *p* < 0.05; ** *p* < 0.01; *** *p* < 0.001 comparing saline and endotoxemia condition. † *p* < 0.05, †† *p* < 0.01, ††† *p* < 0.001 compared with sham-treated condition. # *p* < 0.05; ## *p* < 0.01; ### *p* < 0.001 comparing non-survivor endotoxic condition between therapeutic and prophylactic protocol. NS: non-significant. Results are expressed as the mean ± SD (*N* = 12).

**Figure 5 antioxidants-12-00659-f005:**
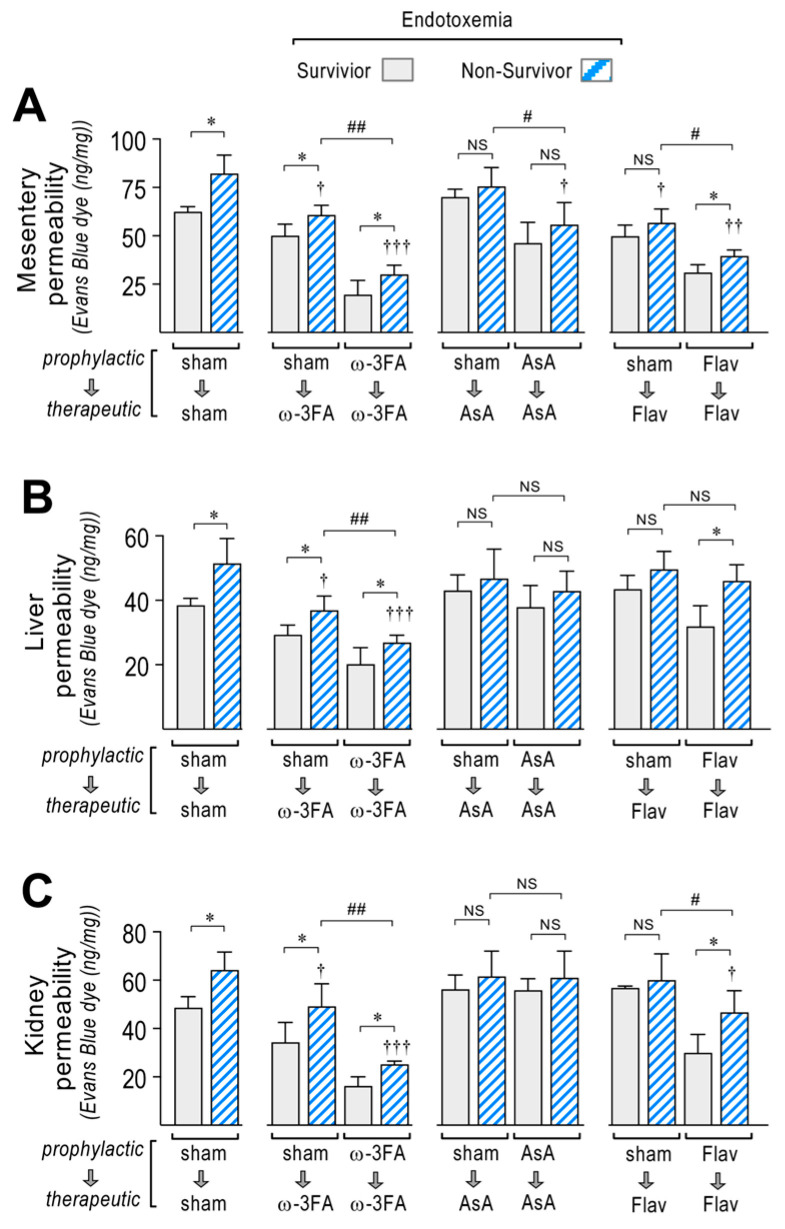
Vascular hyperpermeability through dietary supplementation based on ω-3 FA, ascorbic acid, or polyphenolic antioxidant flavonoids in survivor and non-survivor endotoxemic rats. After endotoxemia treatment supplemented with water (sham), ω-3 fatty acid (ω-3 FA), ascorbic acid (AsA), or polyphenolic antioxidant flavonoids (Flav), through a therapeutic and a prophylactic protocol, the mesentery (**A**), kidney (**B**), and liver (**C**) were harvested and processed for EBD extraction to determine blood vessel hyperpermeability. EBD content was normalized to total sample weight. Statistical differences were assessed by two-way ANOVA followed by Dunnett’s post hoc test. * *p* < 0.05 comparing saline and endotoxemia condition. † *p* < 0.05, †† *p* < 0.01, ††† *p* < 0.001 compared with sham-treated condition. # *p* <0.05; ## *p* < 0.01 comparing non-survivor endotoxic condition between therapeutic and prophylactic protocol. NS: non-significant. Results are expressed as the mean ± SD (*N* = 12).

**Figure 6 antioxidants-12-00659-f006:**
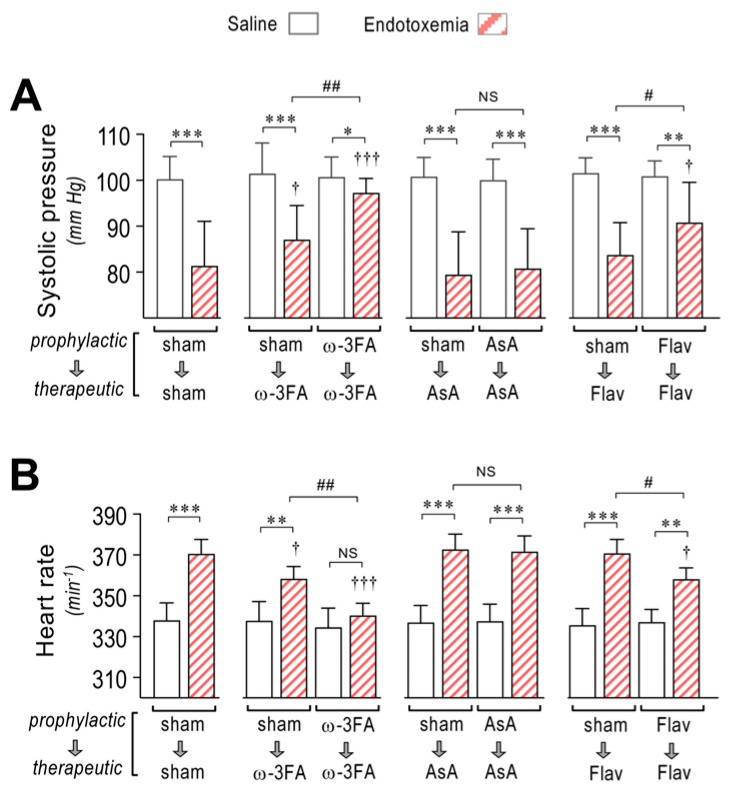
Effects in systolic blood pressure and cardiac frequency through dietary supplementation based on ω-3 FA, ascorbic acid, or polyphenolic antioxidant flavonoids in endotoxemic rats. After saline (open bars) and endotoxemia (dashed bars) treatments that were supplemented with water (sham), ω-3 fatty acid (ω-3 FA), ascorbic acid (AsA), or polyphenolic antioxidant flavonoids (Flav), through a therapeutic and a prophylactic protocol, the systolic blood pressure (**A**) and cardiac frequency (**B**) were determined. Statistical differences were assessed by two-way ANOVA followed by Dunnett’s post hoc test. * *p* < 0.05; ** *p* < 0.01; *** *p* < 0.001 comparing saline and endotoxemia condition. † *p* < 0.05, ††† *p* < 0.001 compared with sham-treated condition. # *p* < 0.05; ## *p* < 0.01 comparing non-survivor endotoxic condition between therapeutic and prophylactic protocol. NS: non-significant. Results are expressed as the mean ± SD (*N* = 12).

**Figure 7 antioxidants-12-00659-f007:**
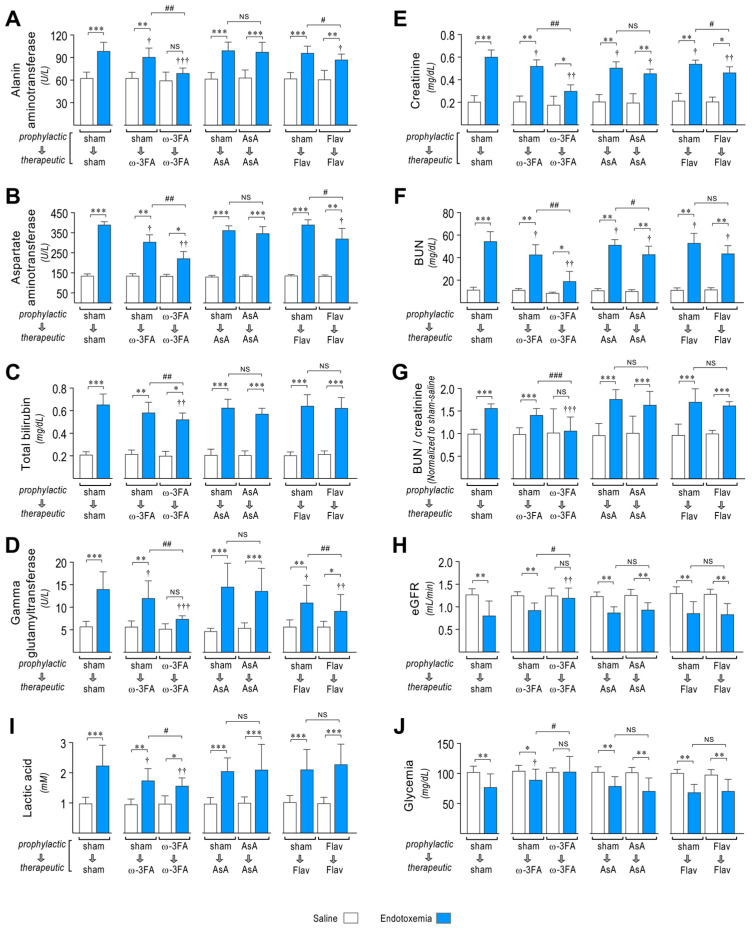
Effects in MODS through dietary supplementation based on ω-3 FA, ascorbic acid, or polyphenolic antioxidant flavonoids in endotoxemic rats. After saline (open bars) and endotoxemia (blue bars) treatments that were supplemented with water (sham), ω-3 fatty acid (ω-3 FA), ascorbic acid (AsA), or polyphenolic antioxidant flavonoids (Flav), through a therapeutic and a prophylactic protocol, the blood levels of the liver dysfunction markers alanine aminotransferase (ALT, (**A**)), aspartate aminotransferase (AST, (**B)**), total bilirubin (TBIL, (**C**)), and gamma-glutamyl transferase (GGT, (**D**)); renal dysfunction markers creatinine (CRE, (**E**)) and blood urea nitrogen (BUN, (**F**)); BUN/Creatinine ratio (**G**); estimated GFR (eGFR, (**H**)); and the metabolic dysfunction markers lactic acid (LAC, (**I**)) and glycemia (GLY, (**J**)) were measured. Statistical differences were assessed by two-way ANOVA followed by Dunnett’s post hoc test. * *p* < 0.05; ** *p* < 0.01; *** *p* < 0.001 comparing saline and endotoxemia condition. † *p* < 0.05, †† *p* < 0.01, ††† *p* < 0.001 compared with sham-treated condition. # *p* < 0.05; ## *p* < 0.01; ### *p* < 0.001 comparing non-survivor endotoxic condition between therapeutic and prophylactic protocol. NS: non-significant. Results are expressed as the mean ± SD (*N* = 12).

**Figure 8 antioxidants-12-00659-f008:**
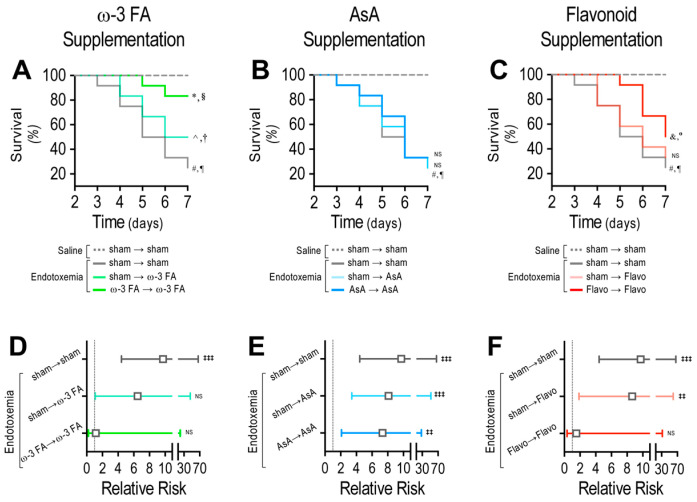
Survival and risk of death through dietary ω-3 FA, ascorbic acid, or polyphenolic antioxidant flavonoid supplementation in endotoxemic rats. Survival (Kaplan–Meier) curves comparing ω-3 FA (**A**), ascorbic acid (**B**), or polyphenolic antioxidant flavonoid (**C**) supplementation in saline- and endotoxemia-treated rats, through a therapeutic and a prophylactic protocol. *, § = 0.01 and 0.03 (log-rank (Mantel–Cox) test and Gehan–Breslow–Wilcoxon test, respectively) when comparing with endotoxemia-treated condition. ^, † = 0.004 and 0.007 (log-rank (Mantel–Cox) test and Gehan–Breslow–Wilcoxon test, respectively), when comparing with endotoxemia-treated condition. #, ¶ = 0.005 and 0.009 (log-rank (Mantel–Cox) test and Gehan–Breslow–Wilcoxon test, respectively), when comparing with saline-treated condition. &, ° = 0.004 and 0.007 (log-rank (Mantel–Cox) test and Gehan–Breslow–Wilcoxon test, respectively), when comparing with endotoxemia-treated condition. NS: non-significant (*N* = 12). Contingency analyses performed to determine relative risk in endotoxemic rats supplemented with ω-3 FA (**D**), ascorbic acid (**E**), or polyphenolic antioxidant flavonoids (**F**) in endotoxemia-treated rats, through a therapeutic and a prophylactic protocol. ‡‡ *p* < 0.01; ‡‡‡ *p* < 0.001 (Fisher’s exact test) compared to saline-treated condition. Results are expressed as the mean ± 95% confidence interval (CI) NS: non-significant (*N* = 12).

## Data Availability

Data is contained within the article and Appendix A.

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
