# Peer review of "Effect of Dietary Supplements with ω-3 Fatty Acids, Ascorbic Acid, and Polyphenolic Antioxidant Flavonoid on Gene Expression, Organ Failure, and Mortality in Endotoxemia-Induced Septic Rats"

_antioxidants, 2023, doi:10.3390/antiox12030659_

Round 1

Reviewer 1 Report

Well-organized and well-written study about the effect of dietary ω-3 fatty acids, ascorbic acid, and polyphenolic antioxidant flavonoid supplements  in endotoxemic-induced septic rats.

I have the following comments:

-please report the number of rats used in each experimental group.

-the authors performed two models: one prophylactic (4 weeks before + the 7 days of endotoxemia) and one therapeutic (the 7 days of endotoxemia). How did they select the time periods of the 4 weeks and the 7 days? (literature, please include ref or preliminary results?)

-I would propose for the authors to make a table where they will present all the supplements given to the animals, the two models (proph+therap) and the conditions examined (gene expression, mortality, etc), where they will add + or  - (effect or no effect). I believe that it will help the readers to have a better and full view of the results.

-Finally, I would like to read a comment about the potential application of these models in ICU patients. For example, what's the meaning of a prophylactic model in these cases, as someone could become septic suddenly and unexpectedly? Concerning the therapeutic model, the rats received the supplements by gavage, how could this be adjusted in patients? (levin, parenteral nutrition etc)

Reviewer 2 Report

General comments
The MS no: manuscript ID _ antioxidants-2239645: "Impact of dietary ω-3 fatty acids, ascorbic acid, and polyphenolic antioxidant flavonoids on gene expression, organ failure, and mortality in endotoxemia-induced septic rats" by authors: Yolanda Prado, Cesar Echeverría, Carmen G. Feijóo, Claudia A. Riedel, Claudio Cabello-Verrugio, Juan F. Santibanez, and Felipe Simon, present a study on the effects of consumption of dietary supplements containing ω-3 fatty acids, ascorbic acid, and polyphenolic antioxidant flavonoids on the modulation of fibrotic endothelium gene expression patterns during sepsis and attempt to determine the effects on sepsis outcomes. The results presented show that the intake of dietary supplements based on ω-3 fatty acids, ascorbic acid, and polyphenolic antioxidant flavonoids was able to reduce the endotoxemia-induced alteration in gene expression associated with endothelial fibrosis. Endotoxemia-induced mortality was reduced by consumption of ω-3 fatty acid-based dietary supplements, mainly by prophylactic intake and therapeutic use, and by consumption of polyphenolic antioxidant flavonoids by prophylactic intake, whereas consumption of ascorbic acid showed no significant difference.

The title of MS is clear and appropriate, but my suggestion is: Effect of dietary supplements with...is clear and appropriate, but my suggestion is: Effect of dietary supplements with...

The abstract is well written, clear, and self-explanatory for readers.

The introductory section is well written and explains the basic concepts of sepsis from cellular to systemic damage with special emphasis on sepsis-induced oxidative stress and the impact of specific antioxidant preparations. The authors explain the role of endothelial dysfunction in the pathogenesis of sepsis syndrome caused by endotoxemia. Special attention is given to the effect of combining antioxidant supplements such as ω-3 fatty acids, ascorbic acid, and polyphenolic antioxidant flavonoids, because there are reports that intake of certain nutrients modulates sepsis. The last paragraph should be changed (explanation in the text).

The Materials and Methods are standard and appropriate. All statistical analyzes are appropriate. Please explain when you used a parametric t-test and when you used a nonparametric U-test and why you combined the two

The Results, both general and specific, are well presented and explained. The figures are clear and representative.

The discussion and conclusions are consistent with the results obtained.

In Conclusion, this MS represents a well-designed and executed experiment with detailed and well presented and written results which represent a contribution to the overall scientific state of the art in this field and provide a solid basis for further research.

Specific comments can be found in the text.

In view of all this, I propose to the editor that this review manuscript be accepted for publication after minor revision.

My final opinion: Suitable for publication.

Round 2

Reviewer 1 Report

The authors addressed all my comments very thoroughly.

I believe that the manuscript is really improved and suitable for publishing.